# Development of a Short-Cut Combined Magnetic Coagulation–Sequence Batch Membrane Bioreactor for Swine Wastewater Treatment

**DOI:** 10.3390/membranes11020083

**Published:** 2021-01-23

**Authors:** Yanlin Chen, Qianwen Sui, Dawei Yu, Libing Zheng, Meixue Chen, Tharindu Ritigala, Yuansong Wei

**Affiliations:** 1State Key Joint Laboratory of Environment Simulation and Pollution Control, Research Center for Eco-Environmental Sciences, Chinese Academy of Sciences, Beijing 100085, China; ylchen_st@rcees.ac.cn (Y.C.); qwsui@rcees.ac.cn (Q.S.); dwyu@rcees.ac.cn (D.Y.); lbzheng@rcees.ac.cn (L.Z.); mxchen@rcees.ac.cn (M.C.); tharindu_st@rcees.ac.cn (T.R.); 2Laboratory of Water Pollution Control Technology, Research Center for Eco-Environmental Sciences, Chinese Academy of Sciences, Beijing 100085, China; 3University of Chinese Academy of Sciences, Beijing 100049, China; 4Institute of Energy, Jiangxi Academy of Sciences, Nanchang 330029, China

**Keywords:** magnetic coagulation, SMBR, nitritation–denitritation, TP, swine wastewater

## Abstract

A high concentration of suspended solids (SS) in swine wastewater reduces the efficiency of the biological treatment process. The current study developed a short-cut combined magnetic coagulation (MC)–sequence batch membrane bioreactor (SMBR) process to treat swine wastewater. Compared with the single SMBR process, the combined process successfully achieved similarly high removal efficiencies of chemical oxygen demand (COD), total nitrogen (TN), ammonium nitrogen (${NH}_{4}^{+}$-N), and total phosphorous (TP) of 96.0%, 97.6%, 99.0%, and 69.1%, respectively, at dosages of 0.5 g/L of poly aluminium chloride (PAC), 2 mg/L of polyacrylamide (PAM), and 1 g/L of magnetic seeds in Stage II, and concentrations of TN, COD, and ${NH}_{4}^{+}$-N in effluent can meet the discharge standards for pollutants for livestock and poultry breeding (GB18596-2001, China). The nitrogen removal loading (NRL) was increased from 0.21 to 0.28 kg/(m^3^·d), and the hydraulic retention time (HRT) was shortened from 5.0 days to 4.3 days. High-throughput sequencing analysis was carried out to investigate microbial community evolution, and the results showed that the relative abundance of ammonia-oxidizing bacteria (AOB) in the SMBR increased from 0.1% without pre-treatment to 1.78% with the pre-treatment of MC.

## 1. Introduction

According to the Bulletin of the Second National Census of Pollution Sources in China, announced in June 2020, the livestock industry is the main source of water pollutants discharged in the agricultural industry, i.e., chemical oxygen demand (COD), ${NH}_{4}^{+}$-N, total nitrogen (TN), and total phosphorous (TP) accounted for 93.76%, 51.30%, 42.14%, and 56.46%, respectively, in 2017 [1]. It was reported that the amount of swine wastewater produced in China increased by approximately 0.16 billion tons every year [2], and swine wastewater treatment is of great concern due to the high concentration of organic matter, nitrogen, phosphorus, suspended solids, and other toxic compounds [3,4,5], such as COD (5512–55,000 mg/L), ${NH}_{4}^{+}$-N (110–1650 mg/L), TN (200–2055 mg/L), TP (100–620 mg/L), and suspended solids (SS) (1117–4837 mg/L) [6,7,8].

Currently, conventional wastewater treatment processes are widely used in the swine wastewater treatment industry, mostly consisting of physical, chemical, and biological treatment processes. These biological treatment processes are combined with an anaerobic digestion process [9] to reduce the concentration of COD, and anoxic/oxic (A/O) processes to remove further dissolved COD and nitrogen in swine wastewater [10]. However, the conventional biological treatment processes are lengthy and require a long hydraulic retention time (HRT), ranging from 10 to 60 d [11] and require a large bioreactor and more land area. Therefore, the high capital, operation, and maintenance costs are considered the main obstacles for treating conventional swine wastewater treatment processes and its spreading applications [12]. A sequence batch membrane bioreactor (SMBR)-based nitritation–denitritation process was developed by Sui et al. to treat swine wastewater, and it was reported that the removal efficiencies of COD, ${NH}_{4}^{+}$-N, and TN were 95%, 99%, and 93%, respectively, and the effluent water quality met the pollutant discharge standard for livestock and poultry breeding (GB18596-2001, China) [13]. The high concentration of SS causes inhibition of microbial communities in subsequent biological treatment and leads to clogging in bioreactors and disintegration of the activated sludge and, finally, low removal efficiency [3,14,15,16]. Therefore, there is a necessity to develop a cost-effective pre-treatment process to remove the SS and improve the performance of subsequent biological treatment processes in swine wastewater treatment.

Coagulation–flocculation is widely used as a pre-treatment process to treat domestic and industrial wastewater, and poly aluminium chloride (PAC)-based coagulation is considered as the most popular coagulation method. However, conventional coagulation is considered a time-consuming process which requires approximately more than 30 min for settling. Additionally, low floc strength, poor density, and loose structure are other drawbacks in conventional coagulation for treating swine wastewater. The magnetic coagulation (MC) process drastically accelerated the sedimentation rate and shortened the HRT by up to 2–5 min [17,18], and reduced the sludge volume by 30% compared with conventional coagulation [17,19]. The MC process also shows advantages such as better removal performance, denser sludge, smaller required land area, and recovery of magnetic seeds [20]. In the conventional coagulation process, insoluble metal phosphate precipitates are formed by hydrolysis of the coagulant, and the precipitates are removed by settling. However, compared with conventional coagulation, magnetic coagulation produces turbulent force after adding magnetic powder, which increases the effective collision between the particles. Magnetic adsorption and aggregation make the magnetic coagulation process more effective than conventional coagulation, improving the structure and strength of the flocs, thereby increasing the efficiency of solid–liquid separation and achieving sludge reduction [19] and removing suspended particles and colloidal substances in wastewater, thus reducing the chemical agent consumption and improving the removal efficiency [21]. As a cost-effective pre-treatment process, magnetic coagulation is being paid more and more attention, and it has been widely applied in high-strength wastewater treatment with high removal efficiencies of COD and TP [19,22] due to its effective removal of SS, COD, phosphate, and TP, which could mitigate the organic loading rate and improve the treatment performance for subsequent biological wastewater treatment processes [22,23].

Though the SMBR showed excellent performances for removing COD, ${NH}_{4}^{+}$-N, and TN [13], the high SS concentration in the swine wastewater may reduce the treatment loading of the SMBR and have impacts on membrane fouling. The SMBR did not include phosphorus removal, and it cannot meet the increasing demands for nutrient removal in the future due to increasingly stringent regulations in China. Therefore, the purpose of this research is to develop a short-cut process by combining MC and an SMBR to meet the discharge requirements of COD and ammonia without carbon addition in treating swine wastewater, and then remove as much phosphorus as possible. In this study, the coupling of swine wastewater pre-treatment under different MC conditions and the continuous operation of the subsequent SMBR process are investigated by lab-scale experiments. The microbial community evolution in the SMBR, as well as the bacteria related to nitrogen transformation, are analyzed by utilizing a high-throughput sequencing analysis.

## 2. Materials and Methods

### 2.1. Setup and Experimental Design of the Combined Process

The combined MC–SMBR process setup consisted of an MC unit and an SMBR unit, as shown in Figure 1. The MC unit was made of cylinder plexiglass with an effective volume of 4 L (diameter 200 mm × height 190 mm), and a mechanical stirring device was installed in the middle of the reactor. The SMBR with an effective volume of 30 L (L 260 mm × W 260 mm × H 450 mm) was equipped with an air compressor (Aerator I) for oxygen supply by aeration, and a mechanical stirrer was placed in the center of the reactor for mixing during the feeding and reaction phases. A polyvinylidene fluoride (PVDF) flat membrane module (SINAP Membrane Tech Co, Shanghai, China) with a pore size of 0.1 mm and a surface area of 0.5 m^2^ fixed in a frame was installed in the back of the reactor [13]. In addition, the perforated pipe with a pore size of 5 mm (Aerator II) and a rate of 5 L/min was set under the membrane module for membrane flushing to control membrane fouling. The operational temperature was held during the experiment between 25 °C and 28 °C using a water bath.

### 2.2. Raw Swine Wastewater and Operational Parameters of Magnetic Coagulation

The raw swine wastewater used in this study was collected from a confined swine farm with a holding capacity of 10,000 heads and stored in a cooler room at 4–6 °C. The characteristics of the raw swine wastewater are listed in Table 1. PAC was used as a coagulant, the flocculant was polyacrylamide (PAM) in the experiments, and magnetic powder was used as an additive to accelerate the settling rate. PAC, PAM, and magnetic seeds were purchased from the commercial market as industrial-grade reagents.

The MC tests were performed batch-wise using a coagulation device, as shown in Figure 1. The steps of the experiments were as follows: (1) at room temperature, 4 L of swine wastewater were added to the MC device; (2) the liquid was rapidly stirred at 300 r/min for 60 s for coagulation, and PAC and magnetic powder (particle size 13 µm) were added immediately after the fast stirring; (3) the mixture was then slowly stirred at 70 r/min for 110 s for flocculation, and polyacrylamide (PAM) was added immediately after this slow stirring; (4) the mixture was allowed to settle for 20 min. After the settling, the supernatant samples were collected in a storage tank for the subsequent SMBR treatment.

To explore the effects of MC on the biological nitrogen removal performance of the SMBR, the experiment was divided into three periods according to the different conditions of the MC, as follows:

Stage I (D1–91): As the control, the swine wastewater was directly fed into the SMBR reactor without MC pre-treatment.

Stage II (D92–135): The swine wastewater was fed into the SMBR reactor after the MC pre-treatment. Dosages of 0.5 g/L of PAC, 2 mg/L of PAM, and 1 g/L of magnetic seeds were used during MC in this stage.

Stage III (D136–171): The swine wastewater was fed into the SMBR reactor after an enhanced MC pre-treatment process to improve phosphorus removal. The enhanced dosages were 1.5 g/L of PAC, 2 mg/L of PAM, and 1 g/L of magnetic seeds.

### 2.3. Operation of the SMBR

According to Sui et al., real-time control logic was applied to the SMBR [13]. At the beginning of each cycle, 1 L of wastewater was pumped into the reactor when it was in the anoxic phase, then the air pump was started to supply oxygen for the oxic phase. The duration of the anoxic and oxic phases was determined by real-time control through online sensors of the pH, Oxidation-Reduction Potential (ORP), and Dissolved Oxygen (DO), indicating the completion of ammonia oxidation and denitrification. Lastly, the membrane efflux pump was started, and the effluent discharge time was set according to the membrane flux and pump power.

The external carbon source (sodium acetate, 10^4^ mg/L) dosing pump in the SMBR reactor was only used in Stage III and not in Stage I and Stage II. The HRT of the SMBR ranged from 4.3 d to 5 d, sludge retention time (SRT) was 15 to 20 d, and the exchange ratio was set at 1/30. The mixed liquor suspended solids (MLSS) in the reactor varied in the range of 6–10 g/L, and the ratio of mixed liquor volatile suspended solids (MLVSS)/MLSS was 0.58–0.77. The loading rates of COD and TN were at 0.2–0.4 kg/(kg·VSS d) and 0.049–0.67 kg/(kg·VSS·d), respectively.

### 2.4. Analysis Methods

The MLSS and MLVSS were determined by the weight method. The concentrations of COD, ${NH}_{4}^{+}$-N, ${NO}_{2}^{-}$-N, ${NO}_{3}^{-}$-N; TN, PO_4_^3−^-P, TP, and SS were analyzed using the standard methods (APHA, 2005). 

Samples of the activated sludge in the SMBR at Stage I (d77, d85, d91), Stage II (d115, d127, d135), and Stage III (d155, d163, d171) were collected for microbial community analysis. DNA extraction was done, as described by Sui et al. [24].

A beaker with a working volume of 1 L was used for the aerobic batch test to evaluate the activities of ammonia-oxidizing bacteria (AOB) and nitrite-oxidizing bacteria (NOB) [25]. The beaker was equipped with a magnetic stirrer, diffuser, and online sensors for monitoring DO, pH, and temperature (Multi 3420, WTW, Munich, Germany). A total of 500 mL of the mixed liquor was collected from the SMBR with a dosage of approximately 40 mg N/L of ammonium for the aerobic batch tests, the temperature was controlled at 25 ± 2 °C, and DO was maintained above 4.0 mg/L. Water samples were collected every 15 min and filtered through a 0.45 µm micropore poly-ethersulfone (PES) membrane to test ammonium, nitrite, and nitrate concentrations. Based on these data, the activities of AOB and NOB were determined by using Appendix A.

### 2.5. Data Analysis

Canoco 5.0 (Microcomputer Power, New York, USA) was used to conduct the principal component analysis (PCA) of the microbial community. The figures were plotted by Origin 9.0 (OriginLab, Northampton, MA, USA). A heatmap was made by using HemI (http://hemi.biocuckoo.org/). The nitrite accumulation ratio (NAR) was calculated using the following equation:(1)$$\mathrm{NAR}=\frac{{\left[{\mathrm{NO}}_{2}^{-}-N\right]}_{\mathrm{eff}}}{{\left[{\mathrm{NO}}_{2}^{-}-N\right]}_{\mathrm{eff}}+{\left[{\mathrm{NO}}_{3}^{-}-N\right]}_{\mathrm{eff}}}\times 100\%$$
where [${\mathrm{NO}}_{2}^{-}$-N]_eff_ (mg/L) and [${\mathrm{NO}}_{3}^{-}$-N]_eff_ (mg/L) are the effluent concentrations of nitrite and nitrate, respectively.

## 3. Results and Discussion

### 3.1. Treatment Performance of the Combined MC-SMBR Process

#### 3.1.1. Pre-Treatment of MC 

The results of the MC process are shown in Figure 2 and Appendix A. The concentrations of COD, ${NH}_{4}^{+}$-N, PO_4_^3−^-P, TP, and SS of influent during Stage II decreased from 10141.3, 1245.7, 140.2, 159.1, and 2263.7 mg/L to 7693.8, 1212.7, 43.4, 52.2, and 1146.4 mg/L, respectively, and the corresponding removal efficiencies are 24.1%, 2.6%, 68.8%, 67.1%, and 49.4%, respectively. The MC exhibited better removal of PO_4_^3−^-P, TP, and SS than COD and ${NH}_{4}^{+}$-N. The C/N ratio of influent was decreased from 8.1 to 5.4, which meets the minimum requirements of the C/N ratio for nitrogen removal by the nitritation–denitritation process.

During Stage III, the concentrations of COD, ${NH}_{4}^{+}$-N, PO_4_^3−^-P, TP, and SS decreased from 11507.3, 1031.3, 134.4, 152.4, and 2375 mg/L to 4936.3, 962.5, 30.5, 35.8, and 271.7 mg/L, respectively, and the corresponding removal efficiencies are 57.1%, 6.7%, 77.3%, 76.5%, and 88.6%, respectively. The C/N ratio of the influent was decreased from 11.2 to 4.1, which means external carbon source dosing was needed in Stage III.

The swine wastewater contains urine, feed residue, flushing water, residual feces, and wastewater produced by livestock and poultry breeding plants [26]. Urine is the primary source of ammonia nitrogen in swine wastewater, and the organic matter comes from residual feces. In the MC process, SS in the wastewater used the magnetic species as the core to form flocs (alum flowers), and flocs were settled with the gravitational force [23]. On the other hand, as an inorganic polymer, PAC could hydrolyze and polymerize to form various aluminum hydroxyl compounds. With a high positive charge and specific surface area, the aluminum hydroxyl compound has a good adsorption effect on impurities and colloidal substances. Simultaneously, it can effectively reduce the zeta potential of colloids and promote the occurrence of flocculation. Al^3+^ also reacts with phosphate ions to form AlPO_4_ to obtain better phosphorus removal efficiency [27]. The addition of magnetic powder is more effective for the adsorption of suspended particles and colloids in wastewater, leading to a better sedimentation performance and higher removal of TP in effluent [21]. The chemical precipitation method based on the MC process was more efficient than other available technologies, such as biological phosphorus removal [28], the adsorption method [29], and struvite crystallization [30] used for phosphorus removal from swine manure. Additionally, MC was not limited to temperature, organic matter, and pH.

The MC method mainly removed the SS, organic matter, and TP from the swine wastewater and increased the corresponding removal along with increasing the PAC dosage. While ensuring the removal efficiency of pollutants, the MC method uses magnetic seeds as the core of the floc, promotes the close combination of flocs, improves the flocculation settling performance, and reduces sludge volume [31]. 

#### 3.1.2. Treatment Performance of the SMBR

The overall performance of the SMBR is shown in Figure 3. In Stage I, swine wastewater without MC pre-treatment was fed into the SMBR. The average concentrations of ${NH}_{4}^{+}$-N, TN, COD, and TP (Appendix A) in the influent were 943.4, 1097.0, 9227.2, and 132.4 mg/L, respectively, and corresponding average effluent concentrations in the SMBR were 8.6, 38.0, 335.9, and 125.3 mg/L, respectively, and the corresponding removal efficiencies of ${NH}_{4}^{+}$-N, TN, COD, and TP were 99.1%, 96.5%, 96.4%, and 5.4%, respectively. The average concentration of nitrite in the effluent reached 13.5 mg/L, and the average NAR was 73.5%. A previous study [13] showed that the hydrogen ions produced during the oxidation of ammonia nitrogen decreased the pH when the ammonia oxidation process was completed; thus, a characteristic “ammonia valley point” appears in the pH variation curve. The distinctive points of the pH change curve can be used to indicate the state of the ammonia oxidation process completion, thus realizing a nitritation–denitritation process to remove nitrogen in the wastewater [32].

In Stage II, with the MC pre-treatment, the average concentrations of ${NH}_{4}^{+}$-N, TN, COD, and TP in the influent were 1212.7, 1422.7, 7693.8, and 52.2 mg/L, respectively, and their average effluent concentrations in the SMBR were 12.4, 36.9, 401.6, and 49.2 mg/L, with removal efficiencies of 99.0%, 97.4%, 94.8%, and 5.7%, respectively. The effluent COD concentration was higher in Stage II compared with Stage I, and the reason was that a high ${NH}_{4}^{+}$-N concentration in the influent caused a high free ammonia concentration (84 mg/L); thus, the activity of COD-degrading heterotrophic bacteria was inhibited. Therefore, it reduced biodegradation due to the inhibition of heterotrophic bacteria and slightly increased the effluent COD concentration in the SMBR. On the other hand, TP removal efficiency was increased to 69.1% through the MC–SMBR process, which is much better than that in the SMBR process without MC pre-treatment. 

Though the influent C/N ratio of the SMBR was reduced from 8.7 to 5.4 due to the MC pre-treatment, the SMBR achieved a better performance of 97.4% of TN removal efficiency without dosing external carbon sources, and the NAR reached 78.1%. The nitrogen removal loading rate increased from 0.21 kg N/(m^3^·d) (Stage I) to 0.28 kg N/(m^3^·d) (Stage II), with an increase of 33.33%. On the one hand, the MC pre-treatment process effectively reduced the influent concentrations of SS, COD, phosphate, and TP. On the other hand, it decreased the HRT from 5.0 d (Stage I) to 4.7 d (Stage II) (Appendix A). Furthermore, the MC promoted the specific activity of AOB (Appendix A) from 5.1 to 10.8 mgN/(mg MLSS·h), decreased the aeration time, and shortened the HRT of the SMBR by 14%. The membrane permeability was increased from 0.23 to 0.28 L/(m^2^·h kPa) (Appendix A), and Transmembrane Pressure (TMP) reduction rates were 0.53, 0.47, and 0.48 kPa/d in Stage I, Stage II, and Stage III, respectively.

In Stage III, the SMBR was operated with enhanced MC pre-treatment for removing more phosphorus. As shown in Figure 3, the average concentrations of ${NH}_{4}^{+}$-N, TN, COD, and TP in the influent were 962.5, 1201.4, 4936.3, and 35.8 mg/L, respectively, and their average effluent concentrations in the SMBR were 7.1, 32.3, 340.3, and 33.2 mg/L, with removal efficiencies of 99.3%, 97.3%, 93.1%, and 7.2%, respectively. The influent C/N ratio of the SMBR further decreased to 4.1 after enhanced MC. An external carbon source (sodium acetate, 10^4^ mg/L) was dosed into the SMBR to ensure the nitrogen removal efficiency of the SMBR during this stage. The average nitrogen removal loading was 0.27 kg N/(m^3^·d), and the HRT was reduced from 4.7 d to 4.3 d, respectively, in Stage II and Stage III (Appendix A). In addition, the nitrite concentration of the effluent averaged at 15.1 mg/L by increasing the contribution of nitrogen removal through a nitration–denitration process. At the end, the NAR reached 78.4%.

In Stage III, the results indicated that with enhanced MC pre-treatment of swine wastewater, most of the TP, PO_4_^3−^-P, SS, and solid organic matter were removed, leading to the residual becoming soluble biodegradable organic matter. Although organic matter is beneficial to improve the utilization efficiency of microorganisms, the influent C/N ratio was too low to guarantee the sufficient required carbon for biological nitrogen removal; hence, an external carbon source was dosed to maintain a high total nitrogen removal efficiency.

Moreover, the combined MC–SMBR process demonstrated excellent removal efficiencies for COD, TN, ${NH}_{4}^{+}$-N, and TP, with removal efficiencies of 97.0%, 97.3%, 99.3%, and 78.2% in Stage III compared with the single SMBR process or other methods (Appendix A) [13,33]. The effluent concentration of COD and ${NH}_{4}^{+}$-N in the combined MC–SMBR process met the limits of COD and ${NH}_{4}^{+}$-N in the Discharge Standard of Pollutants for Livestock and Poultry Breeding of China (GB18596-2001). Additionally, with the MC process, the nitrogen removal loading of the SMBR increased by 33% from 0.21 to 0.28 kg N/(m^3^·d), and the HRT was shortened by 14% from 5.0 d to 4.3 d.

#### 3.1.3. Total Treatment Performance

The short-cut combined MC–SMBR process in Stage II could effectively remove COD, TN, and ${NH}_{4}^{+}$-N, with removal efficiencies of 96.0%, 97.6%, and 99.0%, respectively. The TP removal efficiency was 69.1%, and it was far better than a single SMBR process. Considering the cost of external carbon sources and the required C/N ratio for nitrogen removal in the SMBR, the dosages of PAC, PAM, and magnetic powder at 0.5 g/L, 2 mg/L, and 1 g/L, respectively, were more suitable for swine wastewater pre-treatment.

Compared with a single SMBR process, the combined MC–SMBR process can greatly enhance the solid–liquid separation efficiency and achieve a better removal performance of COD, ${NH}_{4}^{+}$-N, TN, and TP. As a pre-treatment, the MC process effectively decreased the organic load for the subsequent biological treatment and increased the load of biological nitrogen removal. Additionally, it improved AOB activity and the treatment efficiency of the SMBR. In technical and economic aspects, the combined MC–SMBR process could reduce the HRT of SMBR by 14%, thus saving capital and operation and maintenance costs for swine wastewater treatment due to less aeration, a smaller reactor volume, and a smaller land area for biological treatment.

### 3.2. Evolution of Nitrogen Compounds in Typical Cycles of the SMBR

The variation of nitrogen compounds within a cycle at the different stages of the SMBR showed similar behavior, as shown in Figure 4. In Stage I, it can be seen that the SMBR was in an anoxic stage from 0 to 70 min, and concentrations of ${NO}_{2}^{-}$-N and ${NO}_{3}^{-}$-N in the SMBR decreased sharply during the first 30 min of the cycle. The microorganisms used the organic matter in the influent were the electron donors to reduce ${NO}_{2}^{-}$-N and ${NO}_{3}^{-}$-N to nitrogen to achieve nitrogen removal. From the 70th minute, the ${NH}_{4}^{+}$-N concentration decreased from 24.99 mg/L to 16.89 mg/L, and ${NO}_{2}^{-}$-N and ${NO}_{3}^{-}$-N concentrations increased to 13.7 mg/L and 3.95 mg/L, respectively. During this process, the DO gradually increased to approximately 3.8 mg/L, and the pH gradually decreased from 8.19 to 7.89. The “ammonia valley point” appeared at 210 min, and delayed aeration enhanced the removal of organic matter in the swine wastewater. At the end of the cycle, the NAR was 77.6%.

The real-time control strategy was used in the SMBR, in which the “ammonia valley point” on the pH curve was used to judge the end of the ammonia oxidation process to timely stop the aeration. Hence, the nitrification reaction stopped during the ammonia oxidation stage to reach a high NAR. When the ${NH}_{4}^{+}$-N was completely degraded, the “ammonia valley point” appeared on the pH curve, and then ${NO}_{2}^{-}$-N in the reactor was gradually transformed into ${NO}_{3}^{-}$-N. This phenomenon indicated that if the aeration was not stopped in time after the complete ammonia oxidation, the concentration of DO in the reactor might continue to increase, resulting in a decrease in the existing accumulation of ${NO}_{2}^{-}$-N. Therefore, the real-time control strategy was adopted to stop the aeration immediately after the degradation of ${NH}_{4}^{+}$-N in each cycle to prevent further nitrite consumption. After using this real-time aeration strategy, AOB gradually became the dominant nitrifying bacteria, and the NAR gradually increased [34,35].

The total cycle time of all stages was different, i.e., Stage I was 240 min, Stage II was 230 min, and Stage III was 205 min. Therefore, the footprint of the SMBR was reduced by 4% and 14% in Stage II and Stage III, respectively, compared with that in Stage I. As shown in Appendix A, the AOB activities were 5.1, 8.1, and 10.8 mgN/(mg MLSS·h) in Stage I to Stage III, respectively, and the corresponding activity of NOB in Stage I was 1.95 mg/(L·h), and no activity was detected during Stage II and Stage III. The time required for the ammonia oxidation process gradually decreased by 20%, and the HRT of the SMBR decreased from 5.0 d to 4.3 d due to the increased AOB activity. 

### 3.3. Microbial Community Evolution in the SMBR

The microbial communities of activated sludge in the SMBR during Stage I (d77, d85, d91), Stage II (d115, d127, d135), and Stage III (d155, d163, d171) were investigated using a high-throughput sequencing analysis. The relative taxonomic abundances of the microbial communities at a phylum level are shown in Figure 5a. The predominant phyla were *Bacteroidetes* (18.89–72.35%), *Proteobacteria* (6.43–36.25%), *Patescibacteria* (11.25–19.33%), *Firmicutes* (4.21–16.40%), *Chloroflexi* (0.54–14.72%), and *Actinobacteria* (0.83–11.42%). Compared with Stage I, the abundance of *Bacteroidetes* and *Firmicutes* increased during Stage II and Stage III, whereas the abundance of *Proteobacteria*, *Chloroflexi*, and *Patescibacteria* decreased. *Bacteroidetes* was the most abundant phylum and has the ability to metabolize the EPS generated by nitrifiers and secondary metabolites produced by biomass decay [36]. The *Proteobacteria* phylum was correlated with ${NH}_{4}^{+}$-N removal and nitritation efficiency, including the genus *Nitrosomonas*, with high abundances in the SMBR [37,38]. *Chloroflexi* was involved in organic matter degradation, such as polysaccharides [39]. The main phylum level changed, as well as the stable performance of the SMBR, suggesting that there was a close association between the functional bacteria and the pollutant removal efficiency of the SMBR.

The genus-level distributions, abundances, and changes in the AOB and NOB reads during the different stages are shown in Figure 5c. *Nitrosomonas* was detected as the AOB, whereas *Nitrolance* and *Nitrospira* were the NOB. According to the ammonia valley, the relative abundance of AOB in the SMBR was only 0.1% in Stage I due to the control of the oxic phase. After combining with the MC process, the relative abundance of AOB was increased to 1.78%, with a higher growth rate. The reason was that the MC pre-treatment reduced the organic carbon in the influent, which was beneficial for the growth of autotrophic bacteria, such as AOB and NOB. Besides, the reasonable aeration time of the SMBR was conducive to the elimination of NOB.

A redundancy analysis (RDA) was conducted to reveal the species distributions in the samples along with controlling parameters at the genus level (Figure 6). Sludge samples from Stage I, Stage II, and Stage III were clustered in different groups (Appendix A). Sludge samples in Stage I had a positive correlation with the COD/TN ratio of the influent. Sludge samples on days 155, 163, and 171 deviated from the other samples, and positively correlated with the ${NO}_{3}^{-}$-N concentration of the effluent. Distributions of AOB and NOB are described in Figure 6, which are negatively correlated with the COD/TN ratio and predominated in Stage II and Stage III. The abundance of AOB was enriched in Stage II and Stage III due to the removal of organic matter and SS with the introduction of the MC pre-treatment process.

## 4. Conclusions

The short-cut combined MC–SMBR process was successfully developed to treat the swine wastewater, and its removal efficiencies were 99.0%, 97.6%, 96.0%, and 69.1% for ${NH}_{4}^{+}$-N, TN, COD, and TP, respectively. The nitrogen removal loading of the SMBR increased (from 0.21 to 0.28 kg/(m^3^·d)), and the HRT was shortened by 14% (from 5.0 d to 4.3 d) with the introduction of the MC process. Besides, the relative abundance of AOB in the SMBR increased from 0.1% in Stage I to 1.78% in Stage III, which is beneficial for improving AOB activity in the SMBR due to the removal of SS and organic matter by the MC pre-treatment. Considering the economic, technical, and performance factors, the MC in Stage II with dosages of PAC, PAM, and magnetic seeds of 0.5 g/L, 2 mg/L, and 1 g/L, respectively, was more suitable for swine wastewater pre-treatment.

## Figures and Tables

**Figure 1 membranes-11-00083-f001:**
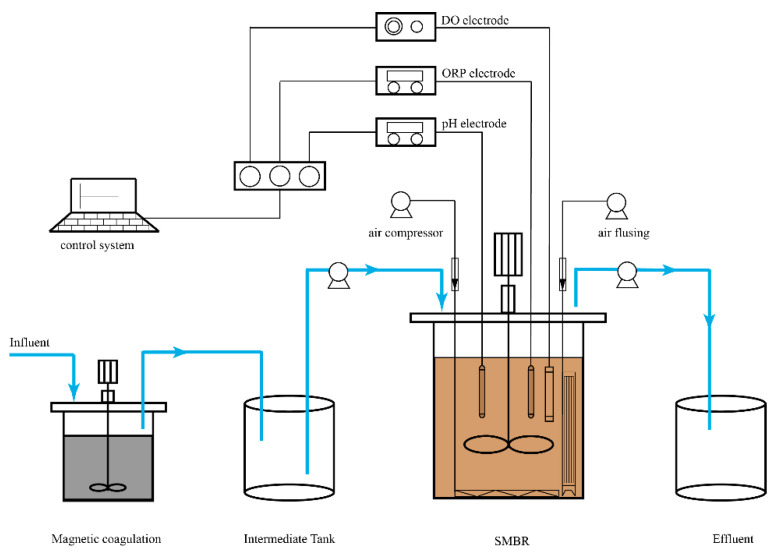
Setup of the combined magnetic coagulation and the sequence batch membrane bioreactor (SMBR).

**Figure 2 membranes-11-00083-f002:**
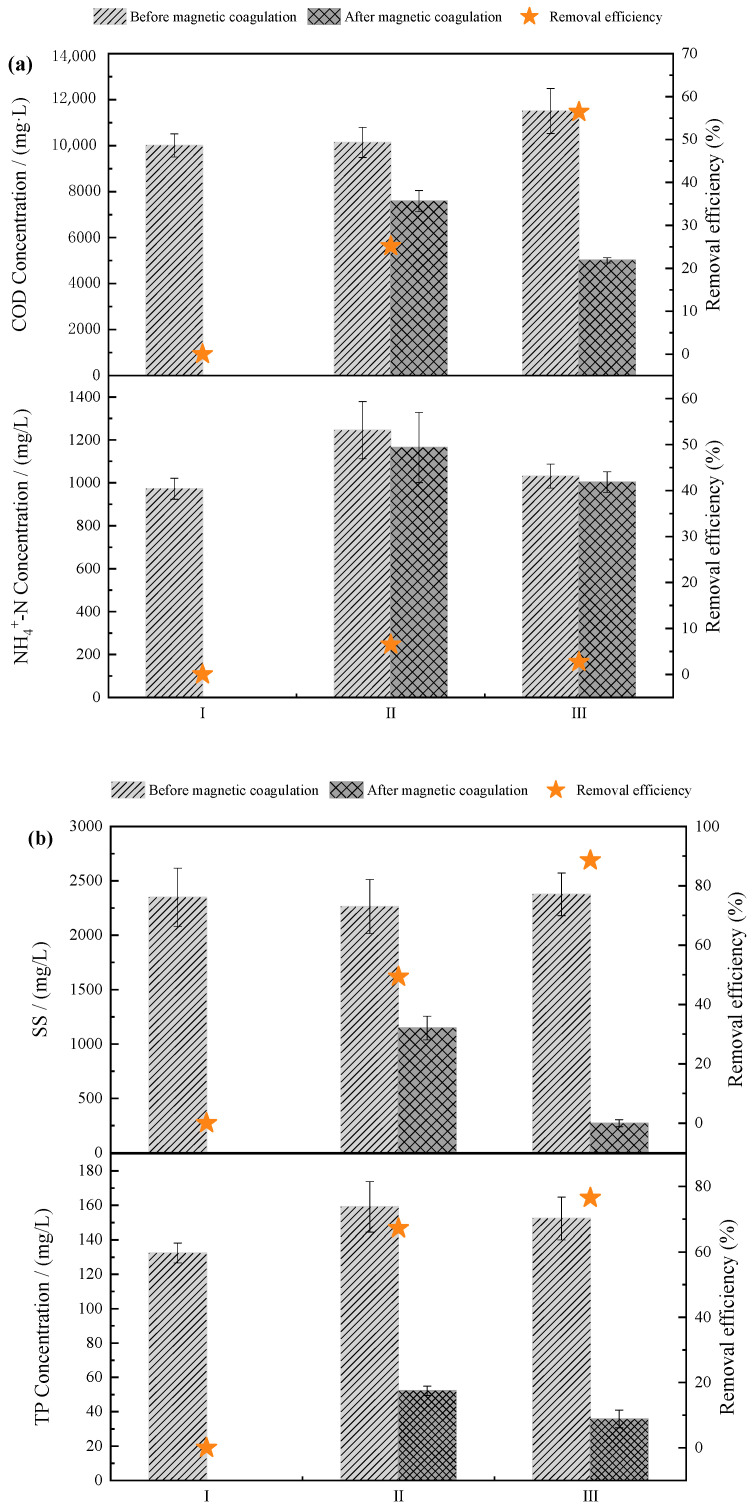
Performance of the magnetic coagulation pre-treatment of swine wastewater. (**a**) COD, ${NH}_{4}^{+}$-N removal performance (**b**) SS, TP removal performance

**Figure 3 membranes-11-00083-f003:**
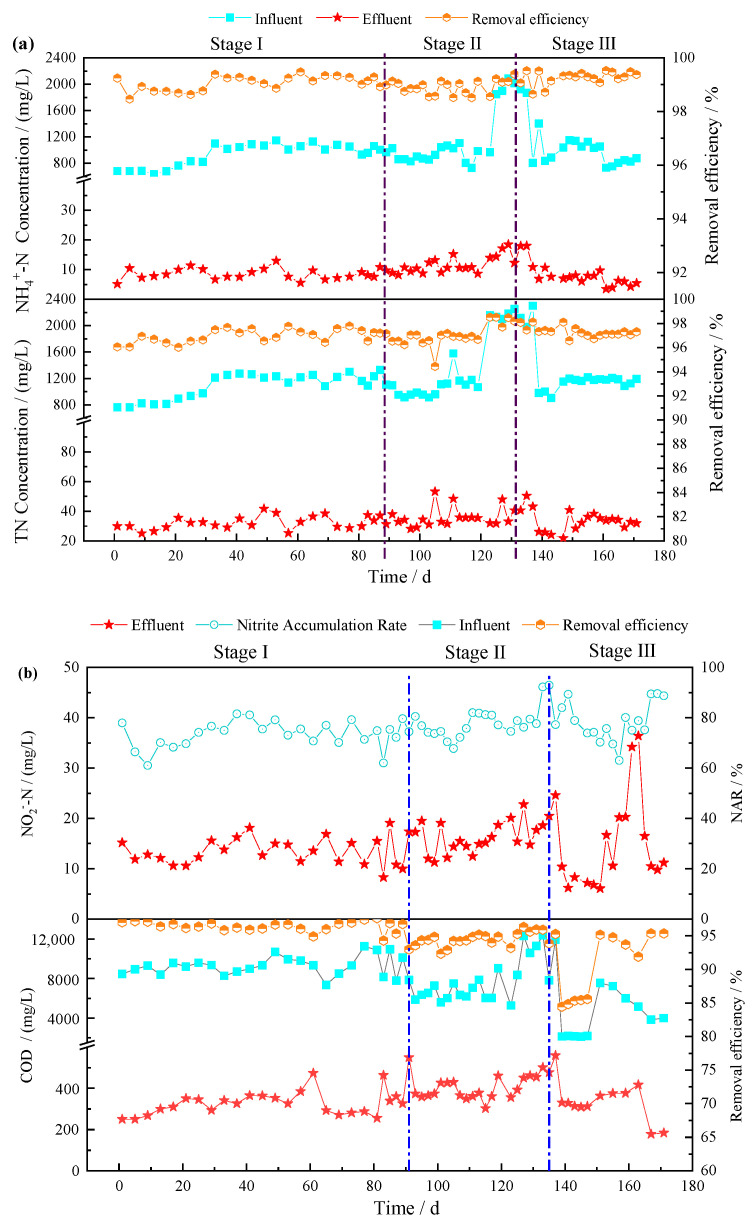
Performance of the SMBR treatment of swine wastewater. (**a**) ${NH}_{4}^{+}$-N and TN removal (**b**) Nitrite accumulation and COD removal

**Figure 4 membranes-11-00083-f004:**
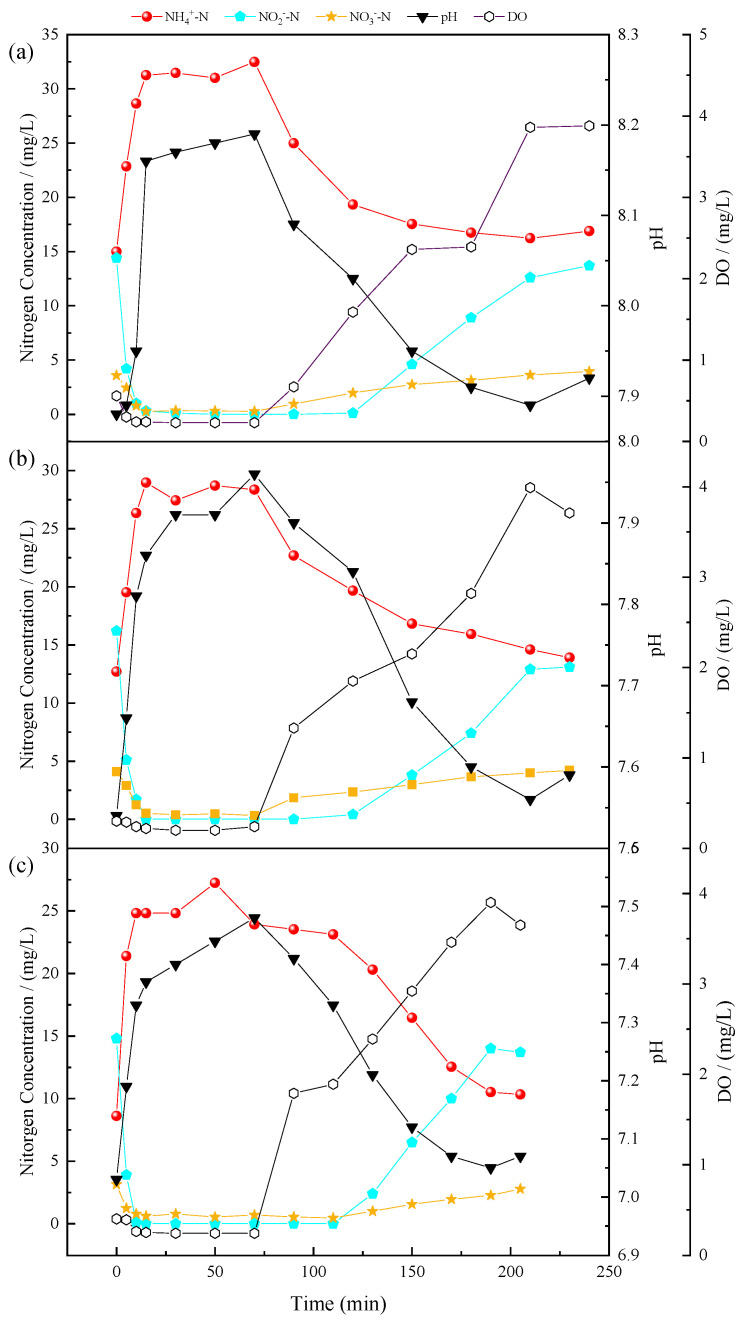
Evolution of nitrogen compounds during cycles in (**a**) Stage I, (**b**) Stage II, (**c**) Stage III.

**Figure 5 membranes-11-00083-f005:**
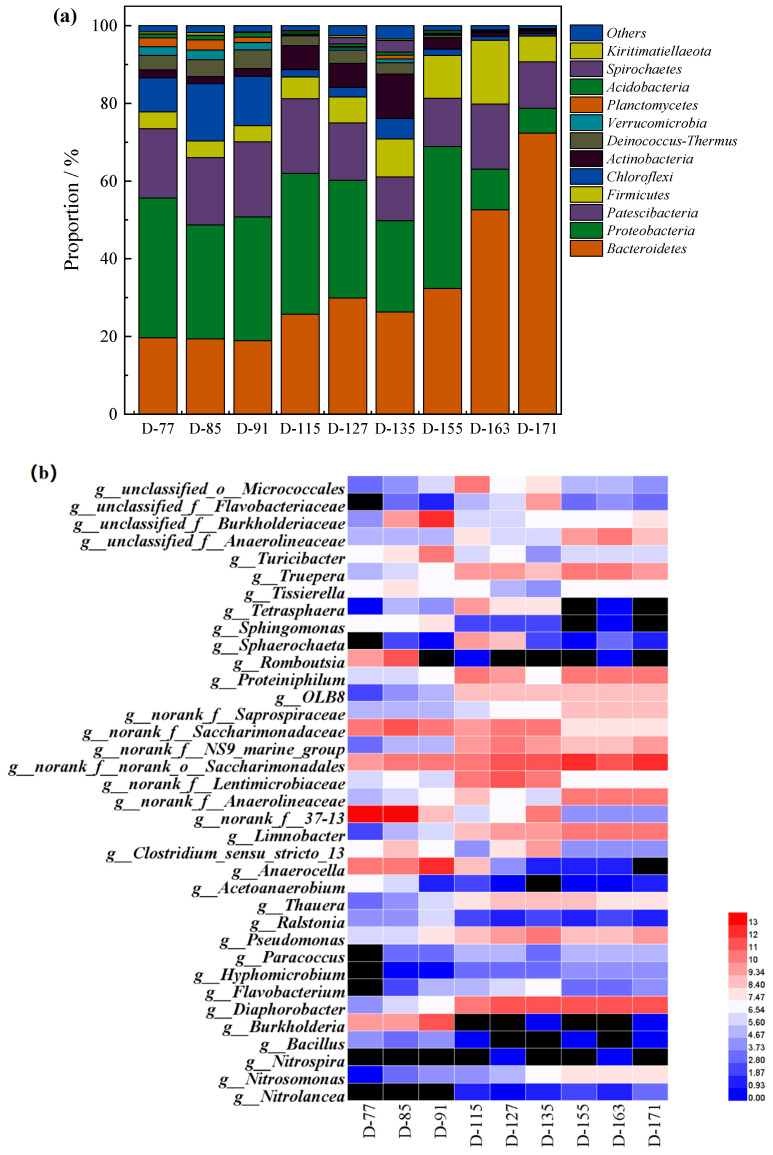
The distribution of bacterial communities in SMBR (**a**) phylum level (**b**) genus level (**c**) Ammonia Oxidizing Bacteria and Nitrite Oxidizing Bacteria.

**Figure 6 membranes-11-00083-f006:**
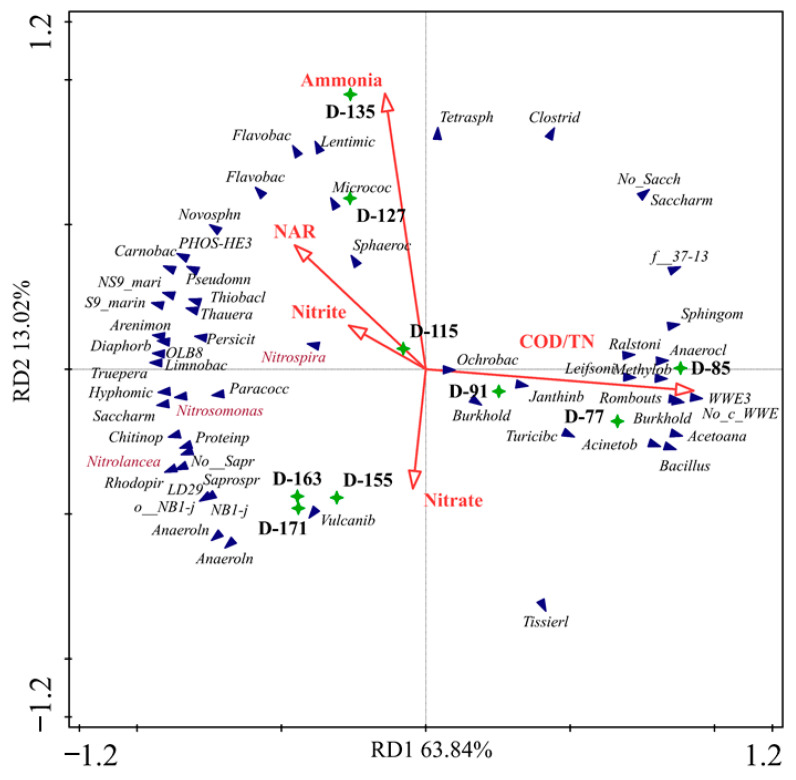
The redundancy analysis (RDA) of the parameters.

**Table 1 membranes-11-00083-t001:** Characteristics of influent swine wastewater at different stages (mg/L).

Stage	COD	NH_4_^+^-N	TN	PO_3_^−^-P	TP	SS	pH	T (°C)
I	9227.2 ± 1012.8	943.4 ± 162.5	1097.0 ± 184.5	114.3 ± 8.6	132.4 ± 5.8	2348.33 ± 268.3	8.1–8.5	25–27
II	10,141.3 ± 1292.3	1245.7 ± 271.2	1514.9 ± 126.4	140.2 ± 15.2	159.1 ± 14.7	2263.7 ± 248.4	8.0–8.5	25–27
III	11,507.3 ± 1817.4	1031.3 ± 181.1	1201.5 ± 145.7	134.4 ± 13.9	152.4 ± 12.4	2375 ± 196.3	8.1–8.4	25–28

## Data Availability

Data is contained within the article or Appendix A.

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
