# Peer review of "Development of a Short-Cut Combined Magnetic Coagulation–Sequence Batch Membrane Bioreactor for Swine Wastewater Treatment"

_membranes, 2021, doi:10.3390/membranes11020083_

Round 1
Reviewer 1 Report
The paper is dealing with a very important environmental concern not only in China but everywhere when intensive livestock units are built in area without enough agricultural land for manure to be spread. The experiments are well described, the results are clear and this paper is useful to better understand the SMBR in relation with carbon and nitrogen load. Because at the end the MC coagulation is useful mainly for P removal, some lieterature about technologies for P removal in manure could be mentionned.
Discussion in the results and discussion part should only discuss about results (see my comments in text)
For example to prove the interest of the magnetic powder an experiment with coagulation only could have been tested.
The advantages of using chemicals against membrane cleaning have to be discussed in regrad with economical end environmental impact of the MC.
The quality of the sludge after the MC depending on the further treatment steps and the possibility for the crops to use the phosphorus when it is trapped with aluminium if the sludge are soread on agricultural lands has to be discussed.

Reviewer 2 Report
This work addressed the treatment of swine wastewater by a magnetic coagulation-sequencing batch membrane bioreactor. High COD, TN and NH4 removal were attained. Overall, the results are interesting. English language can be improved in many parts. Some comments can be found as follows:
Stage II should be defined in the abstract.
Introduction should cover more literature studies on the topic.
What is the intermediate tank for?
Equation 1, used to calculate the nitrite accumulation rate does not take into account the ammonium removed. Why?
Why the DO concentration was so low in the beginning of the cycle? The anoxic/oxic profile was not explained in the materials and methods section.
Figure 5a is hard to follow.
Table S1 can be added to the main manuscript.
Round 2
Reviewer 2 Report
I am satisfied with the responses and changes to the manuscript.
Kind regards